# Conflict violence reduction and pregnancy outcomes: A regression discontinuity design in Colombia

**Giancarlo Buitrago**[1,2]*, **Rodrigo Moreno-Serra**[3]

**1** Clinical Research Institute, School of Medicine, Universidad Nacional de Colombia, Bogota, Colombia,
**2** Hospital Universitario Nacional de Colombia, Bogota, Colombia, **3** Centre for Health Economics, University of York, Heslington, York, United Kingdom

* gbuitragog@unal.edu.co

**Data Availability Statement:** The following information sources: Single Registry of Enrollees, Module ND (Registro Único de Afiliación, or RUAF), Unique Affiliation Database (Base de Datos

## Abstract

### Background

The relationship between exposure to conflict violence during pregnancy and the risks of miscarriage, stillbirth, and perinatal mortality has not been studied empirically using rigorous methods and appropriate data. We investigated the association between reduced exposure to conflict violence during pregnancy and the risks of adverse pregnancy outcomes in Colombia.

### Methods and findings

We adopted a regression discontinuity (RD) design using the July 20, 2015 cease-fire declared during the Colombian peace process as an exogenous discontinuous change in exposure to conflict events during pregnancy, comparing women with conception dates before and after the cease-fire date. We constructed the cohorts of all pregnant women in Colombia for each day between January 1, 2013 and December 31, 2017 using birth and death certificates. A total of 3,254,696 women were followed until the end of pregnancy. We measured conflict exposure as the total number of conflict events that occurred in the municipality where a pregnant woman lived during her pregnancy. We first assessed whether the cease-fire did induce a discontinuous fall in conflict exposure for women with conception dates after the cease-fire to then estimate the association of this reduced exposure with the risks of miscarriage, stillbirth, and perinatal mortality. We found that the July 20, 2015 cease-fire was associated with a reduction of the average number of conflict events (from 2.64 to 2.40) to which women were exposed during pregnancy in their municipalities of residence (mean differences −0.24; 95% confidence interval [CI] −0.35 to −0.13; $p < 0.001$). This association was greater in municipalities where Fuerzas Armadas Revolucionarias de Colombia (FARC) had a greater presence historically. The reduction in average exposure to conflict violence was, in turn, associated with a decrease of 9.53 stillbirths per 1,000 pregnancies (95% CI −16.13 to −2.93; $p = 0.005$) for municipalities with total number of FARC-related violent events above the 90th percentile of the distribution of FARC-related conflict events and a decrease of 7.57 stillbirths per 1,000 pregnancies (95% CI −13.14 to −2.00;

Única de Afiliación, or BDUA), and Calculation Study of the Capitation Unit Database (Base del Estudio de Suficiencia de la Unidad Por Capitación, or UPC) are administered by the Colombian Ministry of Health and Social Protection. These databases are freely available upon request to the Technology of the Information and Communication Office of the Colombian Ministry of Health and Social Protection through the e-mail: correo@minsalud.gov.co. The source of information about armed conflict events was the National Center for Historical Memory (Centro Nacional de Memoria Histórica, or CNMH) dataset, which is publicly available at http://micrositios. centrodememoriahistorica.gov.co/observatorio/ portal-de-datos/el-conflicto-en-cifras/#base-de-datos. We downloaded the dataset on December 8th, 2018.

**Funding:** RMS was supported by a research grant from the UK Medical Research Council (MR/ R013667/1). GB was supported by the School of Medicine at Universidad Nacional de Colombia and University of York (CHE Research Fellowship). The funders had no role in study design, data collection and analysis, decision to publish, or preparation of the manuscript.

**Competing interests:** The authors have declared that no competing interests exist.

**Abbreviations:** CI, confidence interval; FARC, Fuerzas Armadas Revolucionarias de Colombia; IRB, institutional review board; LMICs, low- and middle-income countries; NCHM, National Center for Historical Memory; RD, regression discontinuity; STROBE, Strengthening the Reporting of Observational Studies in Epidemiology.

$p$ = 0.01) for municipalities with total number of FARC-related violent events above the 75th percentile of FARC-related events. For perinatal mortality, we found associated reductions of 10.69 (95% CI −18.32 to −3.05; $p$ = 0.01) and 6.86 (95% CI −13.24 to −0.48; $p$ = 0.04) deaths per 1,000 pregnancies for the 2 types of municipalities, respectively. We found no association with miscarriages. Formal tests support the validity of the key RD assumptions in our data, while a battery of sensitivity analyses and falsification tests confirm the robustness of our empirical results. The main limitations of the study are the retrospective nature of the information sources and the potential for conflict exposure misclassification.

## Conclusions

Our study offers evidence that reduced exposure to conflict violence during pregnancy is associated with important (previously unmeasured) benefits in terms of reducing the risk of stillbirth and perinatal deaths. The findings are consistent with such beneficial associations manifesting themselves mainly through reduced violence exposure during the early stages of pregnancy. Beyond the relevance of this evidence for other countries beset by chronic armed conflicts, our results suggest that the fledgling Colombian peace process may be already contributing to better population health.

## Author summary

### Why was this study done?

- Evidence is lacking about the links between exposure to armed conflict during pregnancy and the risks of miscarriage, stillbirth, and perinatal mortality.

- Only 7 observational studies with a high risk of bias have examined the relationship between conflict exposure and miscarriage, stillbirths, or perinatal mortality. These studies found no evidence of causal effects of conflict exposure on any of these pregnancy outcomes.

- There were no published studies examining the relationship between exposure to conflict violence during pregnancy and the risk of miscarriage, stillbirths, or perinatal mortality in the context of a chronic armed conflict.

### What did the researchers do and find?

- We exploited a discontinuity in the intensity of conflict violence across Colombian municipalities, which was generated by a cease-fire during peace talks between the government and Fuerzas Armadas Revolucionarias de Colombia (FARC), as an exogenous change in conflict exposure for pregnant women. This natural experiment allowed us to implement a regression discontinuity (RD) estimation approach to determine the association between reduced exposure to conflict violence and the risks of miscarriage, stillbirth, and perinatal mortality.

- We found that the fall in the intensity of conflict violence to which women were exposed at the beginning of their pregnancies is significantly associated with decreases in the stillbirth and perinatal mortality rates. We found no associations with the risk of miscarriage.

- The estimated reductions in stillbirth and perinatal death rates associated with reduced conflict exposure were larger for women living in municipalities where the intensity of FARC-related violence was historically greater.

## What do these findings mean?

- Our findings highlight how pregnant women are particularly vulnerable amid armed conflicts and indicate the need for countries to take specific actions to protect these women since the early stages of pregnancy.

- For Colombia, our results offer supportive evidence about the potentially large health benefits arising from the peace process, which can be used to inform current discussions around the political sustainability of this process in the next years.

## Introduction

Over 135 million people today need humanitarian assistance, the vast majority due to armed conflicts [1]. These conflicts have disproportionately affected low- and middle-income countries (LMICs) since the mid-20th century, with devastating consequences for health and development [2,3]. There has been much interest in recent years, by researchers and policymakers alike, in understanding how conflicts affect different health outcomes. Pregnant women and children have received special attention for the study of these effects, due to their high vulnerability to conflict violence [2,4,5]. In LMICs, such vulnerability is often compounded by pervasive healthcare inequities that contribute to higher rates of maternal and infant mortality than in richer countries [6,7]. Indeed, several studies have found that exposure to conflict during pregnancy is associated with short- and long-term adverse outcomes for mothers and their live-born children, both in terms of health and human capital, particularly for poor individuals [8–13].

Yet the aforementioned body of evidence suggesting a causal relationship between exposure to conflict violence and pregnancy outcomes is, in general, weakened by great methodological challenges. Since it is impossible to randomize conflict exposure, there are many possible confounders, and good quality information from contexts of armed conflict is scarce. This includes often patchy information about pregnancy histories in many countries, leading most research to focus on pregnancy outcomes related to live births, and, hence, relying on selected samples. A recent systematic review could find only a few analyses of the associations between conflict exposure and miscarriages (2 studies), stillbirth risk (5 studies), and perinatal mortality (2 studies) [14–20]. All these studies are observational and suffer from data and/or methodological limitations that make causal inference difficult [13] (five studies are before-and-after analyses without an adequate statistical control for observable and unobservable confounders [14–17,19]. A further study is a retrospective cohort analysis that includes a multivariate regression model to identify associations [20], whereas another study relies on before-and-after comparisons within a simple fixed effects panel analysis [18]).

Our study provides evidence of the association between conflict exposure and pregnancy outcomes, taking advantage of a natural experiment created by specific features of the conflict and peace process in Colombia. Colombia has endured one of the longest civil conflicts in history, with over 200,000 direct fatalities and millions of nonfatal victims (mainly civilians) during more than 5 decades [21]. In 2012, the government began a period of peace talks with the country's largest rebel guerrilla, the Fuerzas Armadas Revolucionarias de Colombia (FARC). These conversations, known as the Havana talks, concluded in November 2016 with the signing of a definitive peace agreement between FARC and the government. During the Havana talks, FARC declared unilateral cease-fires on 5 occasions (in 2012, 2013, twice in 2014, and in 2015) before a final, bilateral cease-fire was agreed in August 2016. Due to the fact that these cease-fires mandated a cessation of violent activities by FARC specifically, but not by other rebel groups, the actual reduction in violence should have occurred primarily (if not only) in the geographic areas where FARC used to be active. For our empirical purposes, a cease-fire introduces a plausibly exogenous change in exposure to violence for pregnant women. We are then able to exploit this exogenous change, combined with rich information about pregnancy histories, to determine how the risks of miscarriage, stillbirth, and perinatal death are associated with a reduction in the intensity of conflict violence to which a pregnant woman is exposed in her municipality of residence. Our hypothesis is that reductions in the exposure to armed conflict during pregnancy are associated with reductions in the rates of miscarriages, stillbirths, and perinatal mortality.

## Methods

### Ethics

This study was granted institutional review board (IRB) ethical approval by the Research and Institutional Ethics Committee of the School of Medicine of the Pontificia Universidad Javeriana (Minutes No. 10/2018), Colombia. The study protocol is presented in S1 Protocol.

### Pregnancy cohorts and outcomes

This study included all pregnant women in Colombia between January 1, 2013 and December 31, 2017. We constructed daily single pregnancy cohorts for each of the 1,826 days in the study period. We defined cohorts by the probable conception date. All women were followed until pregnancy ended in either a live birth or fetal death. Newborns were followed since birth until their seventh day of life. We constructed each pregnancy cohort based on birth and death certificates from the Ministry of Health's Single Registry of Enrollees. All live births were identified from birth certificates, and all fetal deaths and deaths before the seventh day of life were identified from death certificates. We merged these 2 databases using an anonymized identifier. The probable date of conception was estimated using the difference between the date of delivery (or death in the case of stillbirths) and the gestational age reported in the birth or death certificates. These data sources were provided by the Colombian Ministry of Health to the Clinical Research Institute of Universidad Nacional de Colombia for use in our research (see Section A in S1 Text for a full description of each information source and the construction of the final database).

Since competing risks of loss in early pregnancy may affect pregnancy outcomes, we followed recent recommendations and used perinatal mortality as our main outcome, defined as death at any point between week 22 of pregnancy (day 154) and the seventh day after birth [22]. We were able to examine fetal deaths also in terms of stillbirth (fetal death after 22 weeks of pregnancy) and miscarriage (fetal death before week 22 of pregnancy) [23]. This study is

reported as per the Strengthening the Reporting of Observational Studies in Epidemiology (STROBE) guidelines (S1 Checklist).

## Exposure to conflict violence

Our measure of individual exposure to conflict violence is the total number of conflict events that occurred in a woman's municipality of residence during her pregnancy. Conflict data were obtained from the Memory and Conflict Observatory database (National Center for Historical Memory, NCHM). This database combines information on conflict-related events from several independent sources, being a leading data source for research about the Colombian conflict due to its reliability and completeness [24]. We downloaded the publicly available NCHM dataset from http://centrodememoriahistorica.gov.co in December 2018, including information about all types of conflict-related events (e.g., murders, kidnappings, sexual violence, land mines, and others) by location. We identified all conflict events that occurred on each day, in each Colombian municipality, between 2013 and 2018 (we included 2018 because women whose pregnancy began at the end of 2017 were also potentially exposed to events in 2018; we excluded from the analysis the events for which it was impossible to determine the date or place of occurrence, representing only 1.11% of all events reported by the NCHM).

We determined exposure to conflict violence for each pregnant woman included in our final dataset, based on the occurrence of conflict events in the municipality where the woman resided during her pregnancy. Thus, in our analytical setting, women "exposed to conflict violence" in their municipality of residence may have been exposed to violent events directly (by suffering violence themselves) or indirectly (by witnessing or knowing of acts of violence committed on others), with potential physical and/or psychological harmful consequences. With the daily information about pregnancy cohorts, we could identify the total number of conflict-related events to which each woman was exposed from the beginning to end of each pregnancy. A woman's municipality of residence was presumed to be the one reported on the birth certificate for live births or death certificate for fetal deaths. Wherever possible, we also cross-verified the municipality of residence using official information from health insurance enrolment and healthcare provision databases to identify whether, during pregnancy, the woman made contact with the health system in the municipality that was assigned as that of residence.

## Cease-fires during the Havana talks

FARC were involved in 7 cease-fires during the Havana talks. Only the last 2 cease-fires before the peace accord, on July 20, 2015 and August 29, 2016, were declared for an indefinite period and lasted more than just a few months, holding until the date of the final peace accord itself (Fig B in S1 Text). The July 20, 2015 cease-fire was declared unilaterally by FARC, followed in August 29, 2016 by a definitive cease-fire by the Colombian government, close to the signing of the peace accord. The other 5 cease-fires were very limited in duration (lasting no more than 4 months), and violence returned to the municipalities soon after these ended. Therefore, we focused on the July 20, 2015 cease-fire for our empirical analyses. This is the first cease-fire that can be expected to have introduced a sudden and permanent reduction in the levels of FARC violence across municipalities, since it was the definitive cease-fire declared and adhered to by FARC. Nonetheless, we also assessed whether the August 29, 2016 cease-fire (which involved both the government and FARC) generated a sharp and lasting reduction in violence. The latter analysis was again performed using regression discontinuity (RD) analysis (see Results).

## FARC presence across municipalities

Several groups other than FARC were responsible for armed violence in certain areas during the study period (Fig C in S1 Text). The cease-fires involving FARC could be expected to have affected violence levels mainly in the areas where FARC had more presence historically. To identify those areas, we grouped all municipalities into categories of FARC-related conflict intensity, based on the distribution of the total number of conflict events involving FARC for the years 2000 through to 2017, thus identifying those municipalities with persistent FARC presence over the past 17 years. We created 4 categories of municipalities: (1) M-p90: municipalities with total number of FARC-related violent events above the 90th percentile of the distribution of these events; (2) M-p75: municipalities with total number of FARC-related events above the 75th percentile; (3) M-zero: municipalities with no conflict event recorded during the 2000 to 2017 period (whether involving FARC or not); and (4) M-other: municipalities that did not belong to the other 3 groups (Fig 1). We chose these categories to achieve the objective of identifying municipal heterogeneity with respect to FARC-related violence levels, in order to investigate the relationships between said heterogeneity and the local responses of violence levels to the cease-fire, without compromising sample sizes for estimation purposes.

## Estimation methodology

We adopted a quasi-experimental RD approach, a methodological design that is well suited to permit robust statistical inference in study settings like ours [25,26]. The cease-fires involving FARC can be regarded as introducing exogenous changes (i.e., discontinuities) in the number of conflict events in the municipalities where women lived during their pregnancies. This setting creates a natural experiment allowing the comparison of outcomes between women with conception dates before and after a cease-fire date (the threshold). The sudden, discontinuous change in exposure to conflict violence implies that assignment to different levels of conflict violence is as good as random for women close to either side of the threshold [25,26]. If the underlying assumptions of the RD design are met (which we can test with the data available), we can use the discontinuous change induced by a cease-fire to estimate the association between exposure to violence and pregnancy outcomes [27–29]. We used the conception date for each cohort as the running variable and the date of the July 20, 2015 cease-fire as the discontinuity threshold (Section C in S1 Text).

We implemented our RD approach through a sequence of estimations steps, each of these subject to formal tests to assess result robustness. First, we examined whether the July 20, 2015 cease-fire was in fact associated with a discontinuity in the total number of conflict events to which women were exposed during pregnancy, separately for events involving (a) FARC; and (b) any armed group (not necessarily FARC). We did so for all municipalities and for each of the categories that we created based on FARC-related events, anticipating that municipalities with greater FARC presence (M-p90 and M-p75) would have seen larger changes in violence associated with the July 20, 2015 cease-fire than municipalities where FARC had less/no presence or where there was no conflict violence prior to the cease-fire (M-zero and M-other). We repeated this process for the August 29, 2016 cease-fire.

Second, for the categories of municipalities where we identified a discontinuity in violence (measured by either FARC-related events or total conflict events), we estimated the association between the implementation of the cease-fire and the risks of perinatal mortality, stillbirth, and miscarriage. We performed these estimations using a nonparametric RD approach with optimal bandwidth [30,31]. This estimation strategy involves approximating the regression functions above and below the threshold (i.e., July 20, 2015) by means of weighted polynomial regressions of order 1 (also called local linear regressions), with weights computed by applying

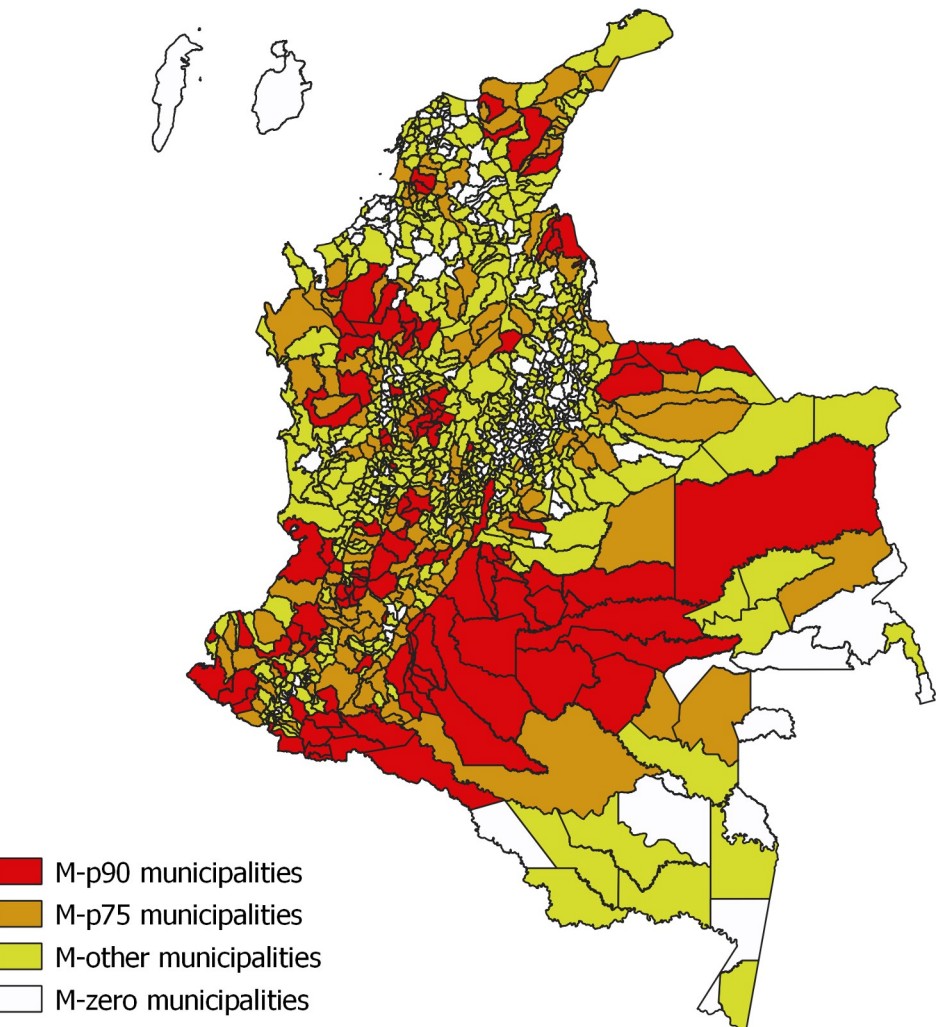

**Fig 1. Distribution of categories of municipalities in Colombia according to FARC presence between 2000 and 2017.** Classification of Colombian municipalities according to the distribution of the total number of conflict events involving FARC for the years 2000 through to 2017. Four categories of municipalities: (1) M-p90: municipalities with total number of FARC-related violent events above the 90th percentile of the distribution of these events (red); (2) M-p75: municipalities with total number of FARC-related events above the 75th percentile (orange); (3) M-zero: municipalities with no conflict event recorded during the entire 2000–2017 period (whether involving FARC or not) (white); and (4) M-other: municipalities that did not belong to any of the other 3 municipality groups (yellow). FARC, Fuerzas Armadas Revolucionarias de Colombia. Base layer of the map from Geoportal DANE (https://geoportal.dane.gov.co/servicios/descarga-y-metadatos/descarga-mgn-marco-geoestadistico-nacional/).

a triangular kernel function on the distance (in days) of each observation to the threshold [32–34]. Similarly to a randomized experiment with imperfect adherence, exposure to conflict took a nondeterministic form (women exposed and not exposed to conflict were on both sides of the threshold) [25]. The parameter of interest was the intention-to-treat estimate.

Third, we tested the validity of the basic RD assumptions. The RD estimates could be biased if the key assumptions of no-manipulation and continuity of baseline variables do not hold in our data. We evaluated the no-manipulation assumption graphically and statistically (McCrary tests) [27]. We also tested statistically for the presence of any discontinuities in several observable baseline characteristics between women with conception dates before and after the cease-

fire. These key baseline characteristics (extracted from the birth or death certificates) included age, education level, health insurance status, marital status, and number of children.

Lastly, we conducted a battery of sensitivity analyses and falsification tests for our main results. We reran all our RD estimations using alternative bandwidths and a parametric approach with first-, second-, and third-order polynomial specifications. The order of the polynomial refers to the highest exponent associated with the running variable of the discontinuous regression (i.e., the distance in days from each observation to the cease-fire date) [27,35]. We also estimated placebo effects by reestimating our models using solely the samples of women living in the municipalities where no discontinuities in conflict events were found (M-zero and M-other), and hence where no associations between cease-fire implementation and pregnancy outcomes should be expected.

For all RD estimations, we report *p*-values from local linear regressions with 95% confidence intervals (CIs). The analyses were performed using Stata 15 (StataCorp LLC, College Station, Texas, US).

## Results

A total of 3,661,022 birth certificates and 1,441,140 death certificates were issued between January 2013 and December 2017 in Colombia (Fig A in S1 Text). After identifying single pregnancies with conception dates between January 1, 2013 and December 31, 2017, the total cohort included 3,254,696 pregnant women. Table 1 presents the characteristics of the cohort for the full sample and separately for women in M-p90 and M-p75 municipalities. Women in M-p90 and M-p75 municipalities had higher pregnancy-related mortality rates than the national average.

### Exposure to conflict violence

On average, the women in our study were exposed to 3.76 (95% CI 3.75 to 3.77) conflict events of any type during pregnancy, with the most frequent types of violent events being selective murder (1.53 events occurred during pregnancy, on average) and sexual violence (0.98 events) (Table 2). Women in M-p90 municipalities were exposed to a higher number of total conflict events on average during pregnancy (7.64; 95% CI 7.62 to 7.66) than women in M-p75 municipalities (6.31; 95% CI 6.29 to 6.32) (mean difference 1.34 (95% CI 1.31 to 1.36; $p < 0.001$)).

Exposure to conflict events during pregnancy showed a sustained falling trend throughout the study period, despite a few spikes (Fig D in S1 Text). Instead of relying on visual inspection, we use RD estimation to identify discontinuities in violence intensity around the cease-fires declared during the study period. We found that the July 20, 2015 cease-fire was significantly associated with a reduction in the average number of conflict events to which a woman was exposed during pregnancy, both with respect to FARC-related and total events (i.e., not necessarily FARC related) (Fig 2, Fig E in S1 Text). Specifically, this cease-fire was associated with a decrease in exposure to violence during pregnancy of 0.24 total events (95% CI −0.33 to −0.08; $p < 0.001$) and 0.03 FARC-related events (95% CI −0.05 to −0.01; $p = 0.01$), on average, for women with conception dates after the cease-fire (Fig 2). Moreover, we found that these reductions in conflict exposure associated with the cease-fire occurred only for pregnant women who lived in M-p90 and M-p75 municipalities. By contrast, repeating these analyses for the August 29, 2016 cease-fire reveals no statistically significant changes in exposure to conflict events during pregnancy (Fig 2).

### Associations between reduced conflict exposure and pregnancy outcomes

Fig 3 shows that the decreased exposure to conflict events during pregnancy, associated with the July 20, 2015 cease-fire, was associated with reductions of 9.53 (95% CI −16.13 to −2.93; $p = 0.005$) stillbirths and 10.69 (95% CI −18.32 to −3.05; $p = 0.01$) perinatal deaths per 1,000

**Table 1. Baseline characteristics of all pregnant women in Colombia between January 1, 2013 and December 31, 2017.**

| | Full sample (n = 3,254,696) | M-p90 (n = 1,288,771) | M-p75 (n = 1,732,875) | M-other (n = 1,469,251) | M-zero (n = 52,570) |
|---|---|---|---|---|---|
| **Sociodemographic characteristics** | | | | | |
| Age—mean (SD) | 24.85 (6.60) | 25.32 (6.65) | 25.06 (6.67) | 24.61 (6.51) | 24.67 (6.66) |
| Age—category n (%) | | | | | |
| <18 y | 426,709 (13.11) | 149,135 (11.57) | 219,233 (12.65) | 200,191 (13.63) | 7,285 (13.86) |
| 18 to 34 | 2,521,718 (77.48) | 1,003,727 (77.88) | 1,339,213 (77.28) | 1,142,234 (77.74) | 40,271(76.60) |
| 35 to 39 | 245,862 (7.55) | 109,010 (8.46) | 139,404 (8.04) | 102,496 (6.98) | 3,962 (7.54) |
| >39 y | 60,407 (1.86) | 26,899 (2.09) | 35,025 (2.02) | 24,330 (1.66) | 1,052 (2.00) |
| Health insurance n (%) | | | | | |
| Contributory | 1,390,665 (42.73) | 692,376 (53.72) | 813,811 (46.96) | 560,725 (38.16) | 16,129 (30.68) |
| Subsidized | 1,703,726 (52.35) | 518,915 (40.27) | 824,155 (47.56) | 844,328 (57.47) | 35,243 (67.04) |
| Other | 160,262 (4.92) | 77,458 (6.01) | 94,882 (5.48) | 64,183 (4.37) | 1,197 (2.28) |
| Education n (%) | | | | | |
| Primary or less | 460,928 (14.73) | 141,260 (11.33) | 231,539 (13.91) | 217,215 (15.37) | 12,174 (23.53) |
| Secondary | 731,012 (23.36) | 252,348 (20.25) | 368,319 (22.13) | 349,885 (24.76) | 12,808 (24.76) |
| Higher than secondary | 1,937,422 (61.91) | 852,698 (68.42) | 1,064,527 (63.96) | 846,146 (59.87) | 26,749 (51.71) |
| Married n (%) | 513,775 (15.79) | 219,873 (17.06) | 273,650 (15.79) | 231,274 (15.74) | 8,851 (16.84) |
| Number of children—mean (SD) | 1.83 (1.21) | 1.74 (1.12) | 1.89 (1.20) | 1.86 (1.21) | 2.04 (1.29) |
| Prenatal care—mean (SD) | 6.47 (2.53) | 6.63 (2.63) | 6.45 (2.61) | 6.48 (2.43) | 6.49 (2.45) |
| **Pregnancy outcomes** Rates per 1,000 pregnancies (95% CI) | | | | | |
| Miscarriage | 45.65 | 72.45 | 61.48 | 28.61 | 12.55 |
| | (45.43 to 45.88) | (72.00 to 72.89) | (61.12 to 61.84) | (28.34 to 28.88) | (12.35 to 12.75) |
| Stillbirth | 8.00 | 10.13 | 9.45 | 6.61 | 3.82 |
| | (7.90 to 8.10) | (9.95 to 10.31) | (9.30 to 9.59) | (6.48 to 6.75) | (3.37 to 4.26) |
| Perinatal death | 11.86 | 13.7 | 13.15 | 10.65 | 4.85 |
| | (11.74 to 11.98) | (13.49 to 13.91) | (12.97 to 13.32) | (10.48 to 10.82) | (4.26 to 5.45) |

Data are means (SD). M-p90: pregnant women in municipalities with total number of FARC-related violent events above the 90th percentile of the distribution of these events. M-p75: pregnant women in municipalities with total number of FARC-related violent events above the 75th percentile of the distribution of these events. M-zero: pregnant women in municipalities with no conflict event recorded during the 2000 to 2017 period (whether involving FARC or not). M-other: pregnant women in municipalities that did not belong to any of the other 3 municipality groups. The number (percentage (%)) of missing values for each variable is age = 0 (0.00%), health insurance = 43 (0.00%), education = 125,334 (3.85%), married = 0 (0.00%), number of children = 29 (0.00%), prenatal care utilization = 174,147 (5.35%), and pregnancy outcomes = 0 (0.00%).

95% CI, 95% confidence interval; FARC, Fuerzas Armadas Revolucionarias de Colombia; SD, standard deviation.

pregnancies, for women in M-p90 municipalities. These figures represent relative reductions of 62.32% in stillbirths and 53.69% in perinatal mortality (Table A in S1 Text). For pregnant women in M-p75 municipalities, we estimated associated reductions of 7.57 (95% CI −13.14 to −2.00; $p = 0.01$) stillbirths and 6.86 (95% CI −13.24 to −0.48; $p = 0.04$) perinatal deaths per 1,000 pregnancies, corresponding to relative reductions of 55.09% in stillbirths and 40.19% in perinatal mortality. We did not find statistically significant associations with respect to miscarriages: Point estimates were −8.52 (95% CI −7.47 to 24.52; $p = 0.30$) in M-p90 municipalities and −3.27 (95% CI −9.78 to 16.32; $p = 0.62$) in M-p75 municipalities.

## Validity of the RD assumptions

Formal statistical tests support the validity of the underlying RD assumptions in our data, and hence of the estimated RD associations (Fig F in S1 Text). Both the histograms of the

**Table 2. Exposure to conflict violence events during pregnancy.**

| | Full sample (*n* = 3,254,696) | M-p90 (*n* = 1,288,771) | M-p75 (*n* = 1,732,875) | M-other (*n* = 1,469,251) | M-zero (*n* = 52,570) |
|---|---|---|---|---|---|
| **Total events[1]** | 3.764 (8.555) | 7.644 (11.930) | 6.308 (10.987) | 0.899 (1.851) | NA |
| **Total FARC-related events** | 0.417 (2.730) | 0.922 (4.219) | 0.765 (3.702) | 0.021 (0.210) | NA |
| **Type of conflict event[2]** | | | | | |
| Terrorist attack | 0.030 (0.171) | 0.075 (0.264) | 0.056 (0.231) | 0.000 (0.010) | NA |
| Act of war | 0.414 (1.955) | 0.877 (2.942) | 0.740 (2.624) | 0.044 (0.282) | NA |
| Attack on populations | 0.000 (0.013) | 0.000 (0.020) | 0.000 (0.018) | 0.000 (0.000) | NA |
| Selective murder | 1.535 (3.385) | 3.096 (4.742) | 2.408 (4.279) | 0.561 (1.361) | NA |
| Kidnapping | 0.061 (0.369) | 0.123 (0.539) | 0.102 (0.487) | 0.015 (0.133) | NA |
| Child recruitment | 0.108 (0.433) | 0.249 (0.636) | 0.199 (0.574) | 0.006 (0.082) | NA |
| Massacre | 0.004 (0.065) | 0.004 (0.059) | 0.007 (0.082) | 0.001 (0.040) | NA |
| Forced disappearance | 0.226 (1.497) | 0.438 (2.316) | 0.353 (2.011) | 0.085 (0.394) | NA |
| Damage to property | 0.293 (1.292) | 0.644 (1.946) | 0.523 (1.728) | 0.033 (0.212) | NA |
| Sexual violence | 0.976 (3.807) | 1.891 (4.891) | 1.712 (5.070) | 0.142 (0.641) | NA |
| Land mine | 0.116 (0.973) | 0.247 (1.499) | 0.208 (1.319) | 0.011 (0.150) | NA |

Data are means (SD). M-p90: pregnant women in municipalities with total number of FARC-related violent events above the 90th percentile of the distribution of these events. M-p75: pregnant women in municipalities with total number of FARC-related violent events above the 75th percentile of the distribution of these events. M-zero: pregnant women in municipalities with no conflict event recorded during the 2000 to 2017 period (whether involving FARC or not). M-other: pregnant women in municipalities that did not belong to any of the other 3 municipality groups.

[1]Note: Most of the conflict events taking place after the start of the peace talks with FARC involved the Colombian armed forces, as the latter continued to fight the other rebel armed groups.

[2]Note: See Section B in S1 Text for an official definition of conflict events.

FARC, Fuerzas Armadas Revolucionarias de Colombia; NA, not applicable; SD, standard deviation.

distribution of women according to conception dates (panel A) and the McCrary tests (panel B) showed no evidence of manipulation, i.e., there is no evidence of discontinuities in the density function of conception dates around the cease-fire (threshold). We also conducted graphical and statistical analyses to assess balance in baseline characteristics between pregnant women on each side of the threshold and again found no evidence of discontinuity for any of these variables (Figs G and H in S1 Text). Taken together, the results above offer strong statistical evidence that exposure to different levels of conflict violence during pregnancy is as good as random for women on both sides of the July 20, 2015 cease-fire [25,34].

## Robustness checks

We tested the sensitivity of our main results to using several different combinations of functional forms and bandwidths. We found that our results of statistically significant reductions in both stillbirths and perinatal mortality, associated with the reduction in violence after the July 20, 2015 cease-fire, are robust to all these specification changes (Tables B and C in S1 Text).

We also estimated associations of the July 20, 2015 cease-fire with pregnancy outcomes for women in M-zero and M-other municipalities as placebo tests (as we did not find any discontinuities in exposure to conflict events in those municipalities). We found no evidence of associations for either stillbirths or perinatal mortality (Table 3). All these results provide further reassurance about the validity of our estimates.

## Discussion

The relationship between conflict violence and the health outcomes of civilian populations has been examined from different perspectives by previous studies [36,37]. Nevertheless, the

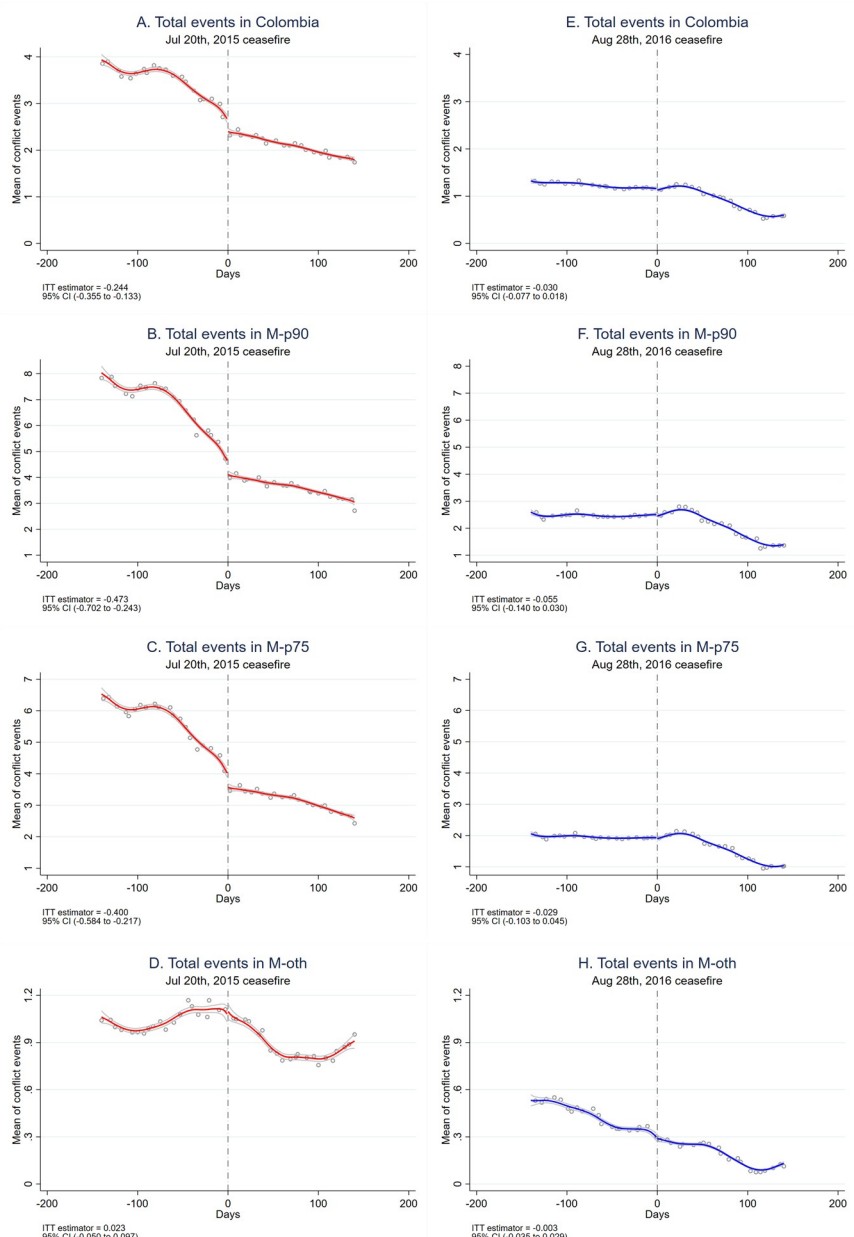

**Fig 2. Association between the July 20, 2015 and August 28, 2016 cease-fires and the exposure to total conflict events during pregnancy: Colombia and categories of municipalities (RD plots).** Association between the July 20, 2015 and August 28, 2016 cease-fires and the exposure to total conflict events during pregnancy (i.e., events related to any armed group). Local linear regressions and 95% confidence interval. A, B, C, and D show the estimates of the July 20, 2015 cease-fire (red lines). E, F, G, and H show the estimates of the August 28, 2016 cease-fire (blue lines). A and E include women from all municipalities in Colombia. B and F only include women in M-p90 municipalities. C and G only include women in M-p75 municipalities. D and F only include women in M-other municipalities. We do not show graphs for women in M-zero municipalities as these women were not exposed to conflict events. M-p90: municipalities with total number of FARC-related violent events above the 90th percentile of the distribution of these events; M-p75: municipalities with total number of FARC-related events above the 75th percentile of the distribution of these events; M-zero: municipalities with no conflict event recorded during the entire 2000 to 2017 period (whether involving FARC or not); and M-other: municipalities that did not belong to any of the other 3 municipality groups. The results indicate that the July 20, 2015 cease-fire was associated with a statistically significant discontinuity in the number of conflict events only for women in M-p90 and M-p75 municipalities, while the August 28, 2016 cease-fire was not associated with in any statistically significant changes in exposure to conflict events for pregnant women. ITT estimates by RD analysis using local linear regression and optimal bandwidth of 56 days. 95% CI, 95% confidence interval; FARC, Fuerzas Armadas Revolucionarias de Colombia; ITT, intention-to-treat; RD, regression discontinuity.

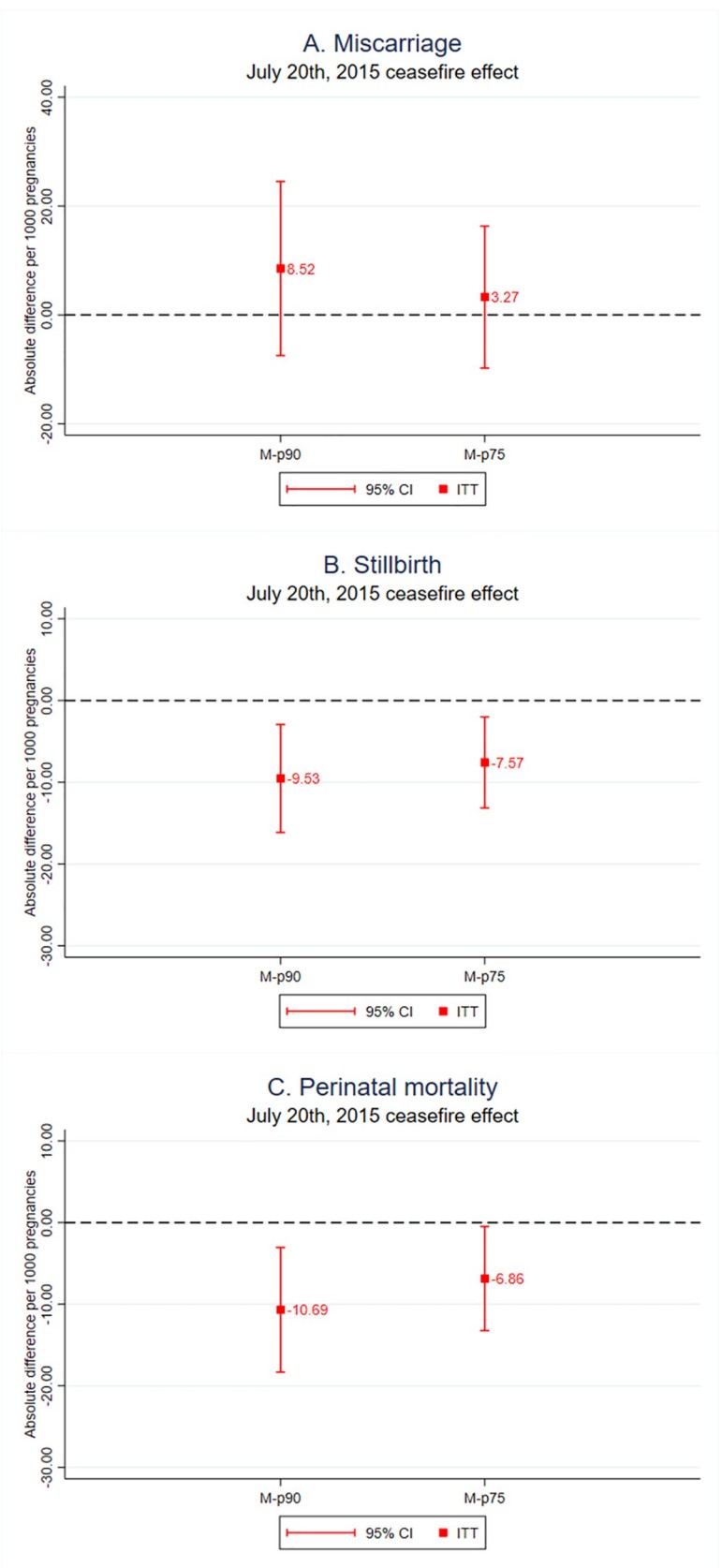

**Fig 3. Association between the July 20, 2015 cease-fire and pregnancy outcomes.** ITT estimates of the July 20, 2015 cease-fire on the risks of miscarriage, stillbirth, and perinatal mortality, for pregnant women in M-p90 and M-p75 municipalities. Estimates obtained through RD analysis using local linear regression and bandwidth of 28 days (Section C in S1 Text). We observe statistically significant estimates on the risk of stillbirths and perinatal mortality, but no effects on the risk of miscarriage. The estimates are larger for women in M-p90 municipalities than for those in M-p75 municipalities. 95% CI, 95% confidence interval; ITT, intention-to-treat; RD, regression discontinuity.

associations between conflict violence and the risks of perinatal mortality and stillbirth have not been evaluated previously with the necessary statistical robustness [13]. Our study employed a RD approach, using cease-fire dates as thresholds that could introduce exogenous discontinuities in conflict intensity in Colombia, to estimate the association between reduced exposure to conflict events during pregnancy and the risks of miscarriage, stillbirth, and perinatal mortality. We found that the July 20, 2015 cease-fire was associated with a decrease in the number of conflict events to which women were exposed during pregnancy, which, in turn, was associated with reduced risks of stillbirth and perinatal mortality.

To our knowledge, our study is the first to robustly estimate the association between exposure to conflict during pregnancy and the risks of stillbirth and perinatal mortality. In addition to methodological shortcomings, data limitations (particularly around pregnancy records) have prevented previous studies from establishing the aforementioned associations. For example, Wagner and colleagues reported that exposure to conflict increased mortality for 1 year olds, 5 year olds, and for reproductive-aged women, as well as the risk of children becoming orphans [5,36]. Further studies have investigated associations between exposure to conflict during pregnancy and other indicators including birth weight, height for age, and cognitive outcomes [8,10,38–40]. Unlike all those previous studies, in our analysis, we were able to reconstruct all pregnancy cohorts in Colombia, since the beginning of each pregnancy, using

**Table 3. Associations between the July 20, 2015 cease-fire and fetal deaths and perinatal mortality.** Placebo tests using M-other and M-zero municipalities.

| | ITT effect | 95% CI | *p*-value | Functional form | Bandwidth | Observations in bandwidth |
|---|---|---|---|---|---|---|
| M-other municipalities | | | | | | |
| Stillbirth | 0.91 | −3.88 to 5.70 | 0.71 | LLR | 28 | 20,833 |
| Stillbirth | 0.69 | −3.23 to 4.61 | 0.73 | LLR | 42 | 31,701 |
| Stillbirth | 0.98 | −2.43 to 4.38 | 0.57 | LLR | 56 | 42,706 |
| Perinatal mortality | 5.16 | −0.89 to 11.21 | 0.09 | LLR | 28 | 20,833 |
| Perinatal mortality | 3.72 | −1.27 to 8.72 | 0.14 | LLR | 42 | 31,701 |
| Perinatal mortality | 3.39 | −0.95 to 7.73 | 0.13 | LLR | 56 | 42,706 |
| M-zero municipalities | | | | | | |
| Stillbirth | 0.06 | −0.06 to 0.18 | 0.32 | LLR | 28 | 778 |
| Stillbirth | −2.52 | −7.46 to 2.42 | 0.32 | LLR | 42 | 1,163 |
| Stillbirth | −3.09 | −9.16 to 2.97 | 0.32 | LLR | 56 | 1,585 |
| Perinatal mortality | 13.26 | −4.10 to 30.61 | 0.13 | LLR | 28 | 778 |
| Perinatal mortality | 9.00 | −5.63 to 23.64 | 0.23 | LLR | 42 | 1,163 |
| Perinatal mortality | 5.52 | −8.91 to 19.94 | 0.45 | LLR | 56 | 1,585 |

** $p < 0.001$, * $p < 0.05$.

Results of the falsification tests using different bandwidths (28, 42, and 56 days) and LLRs We observe no statistically significant estimates of the cease-fire on pregnancy outcomes in M-other and M-zero municipalities (as expected). M-zero: pregnant women in municipalities with no conflict event recorded during the 2000 to 2017 period (whether involving FARC or not). M-other: pregnant women in municipalities that did not belong to the M-p90, M-p75, or M-zero municipality groups. 95% CI, 95% confidence interval; FARC, Fuerzas Armadas Revolucionarias de Colombia; ITT: intention-to-treat effect per 1,000 pregnancies; LLR, local linear regression.

rich administrative data. This enabled us to measure conflict exposure during pregnancy for all women whose pregnancies began between January 1, 2013 and December 31, 2017, following them until the pregnancy ended in fetal death or a live birth. We therefore mitigated the possibility of survival bias influencing our results.

## Strengths and limitations of this study

The validity of our RD approach and its findings are supported by specific historic features of the Colombian peace process. First, while the number of conflict events showed a generally declining trajectory during the entire period of the Havana talks, we found that only the July 20, 2015 cease-fire was associated with a sharp and sustained discontinuity in conflict exposure. This is in line with the fact that for the previous FARC-declared cease-fires, the start/end dates were known and were near in time to one another, and/or the Colombian army continued attacking FARC guerrillas, thus leading to minimal and unsustained changes in violence levels, followed by subsequent rises in these levels (see spikes in violence between 50 and 100 days before the July 20, 2015 cease-fire, Fig E in S1 Text). However, the Havana talks intensified markedly in June to July 2015, leading to the FARC's definitive cease-fire and an accompanying reduction in attacks by the Colombian army against that armed group, in order to support the possibility of a definitive peace agreement. This sharp reduction in violent activities involving the key conflict actors after the July 2015 cease-fire was sustained, resulting in violence levels between FARC and the government that were already very low at the time of the August 29, 2016 bilateral cease-fire.

Second, we found that the July 20, 2015 cease-fire was associated with a higher reduction in conflict violence in municipalities where the FARC had greater presence historically (M-p90) than in municipalities with relatively lower FARC presence (M-p75), whereas no associated changes in violence levels were found in municipalities where FARC were not present or where there was no armed conflict. These (expected) patterns suggest a dose–response relationship between conflict exposure and pregnancy outcomes: The July 20, 2015 cease-fire was associated with greater reductions in stillbirths and perinatal mortality in M-p90 than in M-p75 municipalities and no changes in outcomes in municipalities with no FARC presence or where there were no conflict-related events.

There could have been, of course, several other changes in socioeconomic factors or events (e.g., local health interventions) taking place during the study period, which could influence pregnancy outcomes in general. However, to threaten the validity of our estimates, given the characteristics of our RD estimation design, events, or trends such as those described would need to have been implemented or changed suddenly, discontinuously in the neighborhood of the threshold (i.e., 1 month around the July 20, 2015 cease-fire), and, specifically, for the particular groups of women under comparison (e.g., those living in M-p90 municipalities), in such a way to then introduce discontinuities in the observed distribution of conception dates and/or baseline characteristics at the cease-fire threshold. Yet in our extensive statistical testing of both the no-manipulation and baseline variable continuity assumptions, we found consistent evidence that the distributions of observed conception dates and key observable confounders are continuous at the threshold, thus also supporting the continuity of unobservable and other observable confounders at the threshold [28,29].

The numerous sensitivity tests undertaken indicate that our estimates are robust and provide valuable insights for the interpretation of findings. Although our main estimation results for pregnancy outcomes are based on the same optimal RD bandwidth of 1 month [31], we tested various alternative bandwidth sizes of up to 2 months, with no material changes in our conclusions. Nevertheless, it is noteworthy that the RD point estimates for stillbirths and

perinatal mortality tend to decrease in absolute value as the bandwidth increases from 14 to 56 days (Tables B and C in S1 Text). This suggests that exposure to violence very early in the pregnancy is a key driver of the risks of stillbirth and perinatal mortality.

One limitation of our study is that, with the data available, we were unable to undertake an in-depth investigation of possible mechanisms linking reductions in conflict exposure to reduced risks of stillbirth and perinatal mortality. In this regard, however, evidence from other studies suggest that maternal stress could be a crucial mechanism [20,41–44]. A dysfunction of the hypothalamic–pituitary–adrenal axis (and abnormal cortisol levels) has been identified as the key channel whereby higher levels of stress influence pregnancy outcomes [42,43,45,46]. Simple descriptive analysis of our data offers some support to this possible mechanism. The data on the types of conflict violence to which pregnant women were most vulnerable during the study period indicate that the highest exposure rates were related to very stressful and traumatic events (including events showing a direct relationship with sexual and reproductive health), particularly sexual violence and murder. In addition to reduced risk of suffering traumatic episodes themselves following the July 2015 cease-fire, pregnant women may have also experienced reduced maternal stress stemming from the decrease in violence occurring in the communities where they lived, as such "neighborhood effects" have also been argued to be associated with pregnancy outcomes [47,48]. Related to this, the absence of associations between conflict reduction and miscarriages may be explained by the fact that such fetal losses tend to occur very early in the pregnancy and are mainly caused by chromosomal abnormalities or other factors unrelated to violence exposure during pregnancy [49,50]. Changes in healthcare access due to reduced conflict violence may have played a role for improved pregnancy outcomes as well. However, due to data constraints, we were unable to determine the plausibility of a competing "care access" channel vis-à-vis maternal stress. Although in supplementary analyses we did not find evidence that the reduction in conflict exposure following the July 2015 cease-fire was associated with any changes in prenatal care utilization rates for pregnant women (either in M-p90 or M-p75 municipalities; Table D in S1 Text), we lack suitable data to investigate any other indicators of access to care during pregnancy, which could prove more relevant in the early stages of pregnancy than simple prenatal care utilization (e.g., prenatal care quality, access to essential drugs, or diagnostic tests).

We must note other limitations of this study. First, although Colombia's administrative records have improved substantially in the last decade, some data quality issues may remain and possibly influence some empirical conclusions. One relevant example is that the number of miscarriages reported in our data may well be an underestimation of the true number during the study period, since miscarriages may occur without formal healthcare intervention. Unfortunately, there is no information available on the patterns of underreporting of miscarriages across Colombian municipalities, thus precluding an assessment a priori of whether this phenomenon is likely to have led to an underestimation of our RD associations between conflict exposure and the risk of miscarriage. Yet it does not seem plausible to expect that the July 2015 cease-fire induced sudden changes (e.g., within 1 month) in the patterns of underreporting of miscarriages in M-p90 or M-p75 municipalities, which would be the type of change necessary to influence our estimates (note that a systematic but unchanged pattern of underreporting of miscarriages before and after the cease-fire in a given municipality would not affect the validity of our RD estimates). Data constraints limit our ability to scrutinize this possibility further, however. Second, we measured exposure to conflict based on information about the woman's place of residence taken primarily from birth and death certificates. Although we also conducted cross-checks using other administrative databases wherever feasible, we cannot rule out the possibility that some women migrated, at some point during their pregnancy periods, between municipalities with different intensities of conflict violence. Similarly, we cannot

account in our empirical analysis for potential differences in the intensity of individual conflict exposure, or in the knowledge about conflict events, among pregnant women living in the same municipality. Unfortunately, we do not have detailed geographic information that would allow us to construct finer measures of heterogeneity in individual conflict exposure. Finally, in-depth examination of access and quality of care indicators could have provided valuable insights about possible pathways leading from reduced conflict violence exposure to improved pregnancy outcomes. As previously noted, however, we do not have information about indicators beyond prenatal care use that could be linked up with our dataset of individual pregnancies (such as indicators of quality of prenatal care and access to quality care during labor and delivery or postnatal follow-up for mother and child), and, therefore, we must leave such analyses for future research.

## Conclusions

Our study provides rigorous evidence that reduced exposure to conflict violence during pregnancy is associated with important (yet previously unmeasured) benefits in terms of reduced risks of stillbirth and perinatal mortality. Beyond the relevance of this evidence for other countries beset by chronic armed conflicts, our results offer support to the view that the fledgling Colombian peace process may be already contributing to better population health.

## Supporting information

**S1 Checklist. STROBE Statement.**
(PDF)

**S1 Protocol. Effects of armed conflict during the first three months of life on mortality and the use of health services among infants under 1 year of age.**
(PDF)

**S1 Text. Comprehensive description of methods and complementary analyzes.** Fig A: Flow diagram of the data linkage process. Fig B: Cease-fires declared during the Havana talks. Fig C: Historic presence of armed groups in Colombia (2000 to 2017). Fig D: Trend in the number of conflict events to which pregnant women were exposed during pregnancy in Colombia between January 2013 and December 2017. Fig E: Effects of the July 20, 2015 and August 28, 2016 cease-fires on the exposure to FARC-related conflict events during pregnancy: Colombia and categories of municipalities (RD plots). Fig F: Tests of the RD no-manipulation assumption around the July 20, 2015 cease-fire threshold. Fig G: Tests of balance in baseline characteristics around the July 20, 2015 cease-fire for women in M-p90 municipalities. Fig H: Tests of balance in baseline characteristics around the July 20, 2015 cease-fire for women in M-p75 municipalities. Table A: Effects of the July 20, 2015 cease-fire on fetal deaths and perinatal mortality. Table B: Effects of the July 20, 2015 cease-fire on fetal deaths and perinatal mortality. LLR and parametric regressions by order of polynomial for M-p90 municipalities. Table C: Effects of the July 20, 2015 cease-fire on fetal deaths and perinatal mortality. LLR and parametric regressions by order of polynomial for M-p75 municipalities. FARC, Fuerzas Armadas Revolucionarias de Colombia; LLR, local linear regression; RD, regression discontinuity.
(DOCX)

## Acknowledgments

We thank the Office of Information and Communication Technology of the Colombian Ministry of Health and Social Protection (Dolly Ovalle and Luz Emilse Rincon) for providing the

anonymized data for this study. We would also like to thank Alexander Buritica for his excellent research assistance in the early phases of this project.

## Author Contributions

**Conceptualization:** Giancarlo Buitrago, Rodrigo Moreno-Serra.

**Data curation:** Giancarlo Buitrago.

**Formal analysis:** Giancarlo Buitrago, Rodrigo Moreno-Serra.

**Funding acquisition:** Giancarlo Buitrago, Rodrigo Moreno-Serra.

**Methodology:** Giancarlo Buitrago, Rodrigo Moreno-Serra.

**Resources:** Rodrigo Moreno-Serra.

**Software:** Giancarlo Buitrago.

**Supervision:** Rodrigo Moreno-Serra.

**Visualization:** Giancarlo Buitrago.

**Writing – original draft:** Giancarlo Buitrago.

**Writing – review & editing:** Rodrigo Moreno-Serra.

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
