## [Editor Report · Decision Letter 0]

17 Feb 2021

Dear Dr Buitrago, 

Thank you for submitting your manuscript entitled "The Effects of Conflict Violence Reduction on Pregnancy Outcomes: Evidence from a Regression Discontinuity Design in Colombia" for consideration by PLOS Medicine.

Your manuscript has now been evaluated by the PLOS Medicine editorial staff and I am writing to let you know that we would like to send your submission out for external peer review.

Please re-submit your manuscript within two working days, i.e. by February 19, 2021.

Kind regards,

Beryne Odeny

Associate Editor

PLOS Medicine

---

## [Decision Letter · Decision Letter 1]

12 Apr 2021

Dear Dr. Buitrago,

Thank you very much for submitting your manuscript "The Effects of Conflict Violence Reduction on Pregnancy Outcomes: Evidence from a Regression Discontinuity Design in Colombia" (PMEDICINE-D-21-00816R1) for consideration at PLOS Medicine. We are in receipt of all reviewer comments. As promised, this is an official communication of the major revision decision.

Your paper was evaluated by a senior editor and discussed among all the editors here. It was also sent to four independent reviewers, including a statistical reviewer. The reviews are appended at the bottom of this email and any accompanying reviewer attachments can be seen via the link below:

[LINK]

In light of these reviews, I am afraid that we will not be able to accept the manuscript for publication in the journal in its current form, but we would like to consider a revised version that addresses the reviewers' and editors' comments. Obviously we cannot make any decision about publication until we have seen the revised manuscript and your response, and we plan to seek re-review by one or more of the reviewers. 

We expect to receive your revised manuscript by May 03 2021 11:59PM. Please email us (plosmedicine@plos.org) if you have any questions or concerns.

We look forward to receiving your revised manuscript. 

Sincerely,

Beryne Odeny, 

PLOS Medicine

plosmedicine.org

Please address the following editorial comments:

Please revise your title according to PLOS Medicine's style. Your title must be nondeclarative and not a question. It should begin with main concept if possible. "Effect of" should be used only if causality can be inferred, i.e., for an RCT. Please place the study design ("A regression Discontinuity Design”) in the subtitle (ie, after a colon).

Your study is observational and therefore causality cannot be inferred. Please remove language that implies causality such as “causal evidence” or “causal inference” or “causal effect” or “causes.” Please refer to associations instead.

- The Data Availability Statement (DAS) requires revision. For each data source used in your study:

a) If the data are owned by a third party but freely available upon request, please note this and state the owner of the data set and contact information for data requests (web or email address). Note that a study author cannot be the contact person for the data.

b) If the data are not freely available, please describe briefly the ethical, legal, or contractual restriction that prevents you from sharing it. Please also include an appropriate contact (web or email address) for inquiries (again, this cannot be a study author).

-If you could provide some data to support the validity of their proxy method for disease diagnosis, it would significantly enhance the quality of the manuscript

Please ensure that the study is reported according to the STROBE guideline, and include the completed [STROBE or other] checklist as Supporting Information. When completing the checklist, please use section and paragraph numbers, rather than page numbers. Please add the following statement, or similar, to the Methods: "This study is reported as per the Strengthening the Reporting of Observational Studies in Epidemiology (STROBE) guideline (S1 Checklist).",Please report your study according to the relevant guideline, which can be found here: http://www.equator-network.org/

In the abstract Methods and Findings:

-Please ensure that all numbers presented in the abstract are present and identical to numbers presented in the main manuscript text.

-Please include the actual amounts of relevant outcomes

- Please quantify the main results (with both 95% CIs and p values).

- In the last sentence of the Abstract Methods and Findings section, please describe the main limitation(s) of the study's methodology.

Abstract summary - At this stage, we ask that you include a short, non-technical Author Summary of your research to make findings accessible to a wide audience that includes both scientists and non-scientists. The Author Summary should immediately follow the Abstract in your revised manuscript. This text is subject to editorial change and should be distinct from the scientific abstract. Please see our author guidelines for more information: https://journals.plos.org/plosmedicine/s/revising-your-manuscript#loc-author-summary.

Introduction section: please provide a clear statement of hypothesis at the end of the introduction.

Did your study have a prospective protocol or analysis plan? Please state this (either way) early in the Methods section.

-In statistical methods, please refer to any post-hoc corrections to correct for multiple comparisons during your statistical analyses. If these were not performed please justify the reasons. Please refer to our statistical reporting guidelines for assistance (https://journals.plos.org/plosone/s/submission-guidelines.#loc-statistical-reporting)

In the Methods and Results section:

- Please provide both adjusted and unadjusted estimates.

- Please provide 95% CIs and p values for all HRs.

- When a p value is given, please specify the statistical test used to determine it.

For your figures and tables, please do the following:

- Please define the abbreviations such as FARC, m-P90, m-p75, M-oth, M-zero

Please include line numbers in your next draft.

Comments from the reviewers:

Reviewer #1: This well-written manuscript addresses the important topic of how reductions in conflict violence may affect pregnancy outcomes in low-resource settings with high levels of conflict violence. The authors report on data from over 3 million pregnant women in the country of Columbia between 2013 and 2017. They take advantage of a natural experiment that occurred when ceasefires with FARC occurred in 2015 and 2016 and resulted in dramatic reductions in conflict events. They utilize comprehensive administrative municipality-level data on conflict events that occurred and on adverse pregnancy outcomes (stillbirths, miscarriages, and perinatal mortality) to conduct a rigorous regression discontinuity analysis, and found that decreases in conflict events in municipalities heavily impacted by FARC-related conflict were associated with statistically significant reductions in stillbirths and perinatal mortality (but not miscarriages). The authors have done a good job of describing their methodology, assumptions, sensitivity analyses, and noting the limitations of their research. However, I had a few questions and comments as detailed below. Overall, I thought this was a fascinating analysis and an important contribution to the literature. Specific comments by section are given below.

Abstract:

1. Page 2 and throughout: Given that this necessarily an observational study, it seems like the authors should avoid language such as "the reduction in average exposure to conflict violence resulted in…." throughout the paper. There is not definitive evidence that the reduction in violence caused these outcomes. However, I do recognize that this may be the best evidence we can get on this question.

Introduction:

2. Page 5: Were there other events or trends going on simultaneously in the country during this period that could have also contributed to improved pregnancy outcomes? Campaigns to increase birth in a health facility with a skilled attendant? Increases in maternal education? Changes in the economy?

Methods:

3. Page 10: Which observable baseline characteristics of women before and after the ceasefire were tested statistically? These should be specified. 

Results:

4. Page 11: Based on the characteristics in Table 1, I would like to know if any of the differences in other characteristics of the women are statistically significant among the municipality groups. I think it would be more informative to show all four municipality groups in these tables (not just the total, M-p90, and M-p75), so we can understand more about the differences between those municipalities who were most impacted by FARC-related conflict and those who were less impacted.

Discussion:

5. Page 15: The results of the sensitivity test indicating that exposure very early in pregnancy is the key driver are not explained clearly enough. It is not clear to me why increasing the RD bandwidth to sizes up to two months with no changes in results would lead to this conclusion.

6. Page 16: The discussion of mechanisms of effect is much appreciated. However, could another mechanism be related to improvements in economic activity and the socio-economic status of women due to the environment of reduced conflict? Poverty is associated with adverse health outcomes and perhaps improvements in economic status could be another mechanism.

7. Page 17: Another limitation is that we do not know for certain whether women were directly exposed or even aware of the conflict events that occurred in their municipalities.

Tables and figures:

8. I suggest including all four municipality groups in Tables 1 and 2, for full transparency and information. 

9. Table 2: Please define "Selective murder". Overall, it would be helpful to define all the conflict events in footnotes to the table. What is the difference between a terrorist attack and a massacre?

10. Figure 2: The changing of the scales on the Y-axis for July 2015 and August 2016 is confusing and makes it appear that there were dramatic and similar decreases for both ceasefires. Use of the same scales would make clearer the reason for the lack of effects for the August 2016 ceasefire. This is explained in a footnote, but may be missed by many readers.

Reviewer #2: See attachment

Michael Dewey

Reviewer #3: Article Review 

Thank you for the opportunity to read the manuscript, "The Effects of Conflict Violence Reduction on Pregnancy Outcomes: Evidence from a Regression Discontinuity Design in Colombia". This interesting piece of work uses a quasi-experimental design to explore the causal effect of exposure to violence on pregnancy outcomes. The study benefits from a large sample and an interesting use of administrative data. The sensitivity analyses conducted (e.g., bandwidths and a parametric approach with first-, second-, and third-order polynomial specifications, placebo effects) are a valuable strategy to validate the RD design. 

Abstract

- In the methods and findings paragraph, the authors indicate: "We found that the July 2015 ceasefire reduced the average number of conflict events to which women were exposed during pregnancy in their municipalities of residence (absolute effect -0·20; 95% CI -0·33 to -0·08)". I suggest specifying the effect size index reported here. Is it a difference in means?

Introduction

- Page 4, end of the second paragraph. The authors mention previous research testing the associations between conflict exposure and miscarriages, stillbirth and perinatal mortality. It is stated that all previous evidence is based on observational data. Did any of the studies involve a quasi-experimental design (propensity score matching, interrupted series analysis? If so, I would discuss the limitations of previous quasi-experimental evaluations.

Methods and results

- Please add some details on how the "probable conception date" was calculated. 

- The authors assert that data were extracted from birth/death certificates. However, Table 1 in the findings section describes demographic characteristics of participants such as age, education, marital status, health insurance, number of children, among others. It is not clear if all this information was extracted from the birth/death certificates or from a different source of administrative data. If a different source of data was included in the study, was that data affected by missingness? Did you explore patterns of potential missing data? Please clarify. 

- The description of the RD design and the sensitivity analyses are well developed and offer to the reader enough elements to evaluate the quality of the outcomes.

- It might help to describe in more detail the variables analysed in the study, for example, exposure to conflict events such as selective murder or sexual violence. How are these concepts operationalised?

Discussion

- The discussion on the role of stress as a potential mechanism linking exposure to violence and stillbirth and perinatal mortality deserves more elaboration. Given the aims of the Journal, it would be useful to further elaborate the biological mechanisms underpinning the effects of violence and adversity exposure on maternal and birth outcomes. 

Reviewer #4: Congratulations on both an expertly implemented study using a novel analytical tool (RD) and for writing up your work in an accessible and enjoyable paper. I think this work is publication worthy but there are some minor errors I think need to be attended to. These I have captured on my annotated document attached. Good luck with your publication and follow up studies in this under-researched field.

[LINK]

---

## [Decision Letter · Decision Letter 2]

26 May 2021

Dear Dr. Buitrago,

Thank you very much for re-submitting your manuscript "Conflict Violence Reduction and Pregnancy Outcomes: A Regression Discontinuity Design in Colombia" (PMEDICINE-D-21-00816R2) for review by PLOS Medicine.

I have discussed the paper with my colleagues and the academic editor and it was also seen again by three reviewers. I am pleased to say that provided the remaining editorial and production issues are dealt with we are planning to accept the paper for publication in the journal.

[LINK]

We look forward to receiving the revised manuscript by Jun 02 2021 11:59PM.   

Sincerely,

Beryne Odeny, 

Associate Editor 

PLOS Medicine

plosmedicine.org

Requests from Editors:

1) Thank you for providing your STROBE checklist. Please replace the line numbers with paragraph numbers per section (e.g. "Methods, paragraph 1"), since there will be no line numbers in the final published paper.

2) Please provide the English translation of the Spanish protocol and include it as a Supplementary Information file

3) Please provide p-values for estimates in lines 308-310

4) Please use the "Vancouver" style for reference formatting, and see our website for other reference guidelines https://journals.plos.org/plosmedicine/s/submission-guidelines#loc-references. 

a) Please ensure that weblinks are accessible and dates of access have been updated. For example, the weblink for reference #1 is no longer accessible. 

b) Please provide a weblink for reference #3

Comments from Reviewers:

Reviewer #1: The authors have done an excellent job of responding to all the concerns of the editors and the multiple reviewers. The paper has been considerably strengthened and makes an excellent contribution to the literature. I have no further comments.

Reviewer #2: The authors have addressed my points.

Michael Dewey

[LINK]

---

## [Editor Report · Decision Letter 3]

2 Jun 2021

Dear Dr Buitrago, 

On behalf of my colleagues and the Academic Editor, Dr. Sarah Stock, I am pleased to inform you that we have agreed to publish your manuscript "Conflict Violence Reduction and Pregnancy Outcomes: A Regression Discontinuity Design in Colombia" (PMEDICINE-D-21-00816R3) in PLOS Medicine.

PRESS

Sincerely, 

Beryne Odeny 

Associate Editor 

PLOS Medicine